# Structural determinants of adhesion by Protocadherin-19 and implications for its role in epilepsy

Sharon R Cooper[1,2], James D Jontes[2]*, Marcos Sotomayor[1]*

[1]Department of Chemistry and Biochemistry, The Ohio State University, Columbus, United States; [2]Department of Neuroscience, The Ohio State University, Columbus, United States

**Abstract** Non-clustered δ-protocadherins are homophilic cell adhesion molecules essential for the development of the vertebrate nervous system, as several are closely linked to neurodevelopmental disorders. Mutations in *protocadherin-19* (*PCDH19*) result in a female-limited, infant-onset form of epilepsy (PCDH19-FE). Over 100 mutations in *PCDH19* have been identified in patients with PCDH19-FE, about half of which are missense mutations in the adhesive extracellular domain. Neither the mechanism of homophilic adhesion by PCDH19, nor the biochemical effects of missense mutations are understood. Here we present a crystallographic structure of the minimal adhesive fragment of the zebrafish Pcdh19 extracellular domain. This structure reveals the adhesive interface for Pcdh19, which is broadly relevant to both non-clustered δ and clustered protocadherin subfamilies. In addition, we show that several PCDH19-FE missense mutations localize to the adhesive interface and abolish Pcdh19 adhesion in *in vitro* assays, thus revealing the biochemical basis of their pathogenic effects during brain development.

*For correspondence: jontes.1@osu.edu (JDJ); sotomayor.8@osu.edu (MS)

**Competing interests:** The authors declare that no competing interests exist.

## Introduction

Nervous system function is critically dependent on the underlying neural architecture, including patterns of neuronal connectivity. Cell-cell recognition by cell surface receptors is central to establishing these functional neural circuits during development (*Kiecker and Lumsden, 2005*; *Steinberg, 2007*; *Zipursky and Sanes, 2010*). The cadherin superfamily is a large and diverse family of cell adhesion molecules that are strongly expressed in the developing nervous system (*Hirano and Takeichi, 2012*; *Suzuki, 1996*; *Frank and Kemler, 2002*; *Shapiro et al., 2007*; *Gumbiner, 2005*; *Chen and Maniatis, 2013*). The differential expression of classical cadherins and protocadherins, the largest groups within the cadherin superfamily, suggests that they play important roles in the development of neural circuitry (*Weiner and Jontes, 2013*; *Hirano and Takeichi, 2012*), an idea supported by their involvement in a range of neurodevelopmental disorders (*Redies et al., 2012*; *Hirabayashi and Yagi, 2014*). In particular, the non-clustered δ-protocadherins have been linked to autism spectrum disorders, intellectual disability, congenital microcephaly and epilepsy.

*Protocadherin-19* (*PCDH19*) is a member of the non-clustered δ2-protocadherin subfamily (*Wolverton and Lalande, 2001*; *Vanhalst et al., 2005*; *Gaitan and Bouchard, 2006*; *Emond et al., 2009*; *Liu et al., 2010*) that is located on the X-chromosome. Mutations in *PCDH19* cause an X-linked, female-limited form of infant-onset epilepsy (PCDH19 female epilepsy, PCDH19-FE; OMIM 300088) that is associated with intellectual disability, as well as compulsive or aggressive behavior and autistic features (*Dibbens et al., 2008*; *Scheffer et al., 2008*; *Depienne and LeGuern, 2012*; *van Harssel et al., 2013*; *Leonardi et al., 2014*; *Thiffault et al., 2016*; *Terracciano et al., 2016*; *Walters et al., 2014*). To date, well over 100 distinct mutations in *PCDH19* have been identified in

**eLife digest** As the brain develops, its basic building blocks – cells called neurons – need to form the correct connections with one another in order to give rise to neural circuits. A mistake that leads to the formation of incorrect connections can result in a number of disorders or brain abnormalities.

Proteins called cadherins that are present on the surface of neurons enable them to stick to their correct partners like Velcro. One of these proteins is called Protocadherin-19. However, it was not fully understood how this protein forms an adhesive bond with other Protocadherin-19 molecules, or how some of the proteins within the cadherin family are able to distinguish between one another.

Cooper et al. used X-ray crystallography to visualize the molecular structure of Protocadherin-19 taken from zebrafish in order to better understand the adhesive bond that these proteins form with each other. In addition, the new structure showed the sites of the mutations that cause a form of epilepsy in infant females. From this, Cooper et al. could predict how the mutations would disrupt Protocadherin-19's shape and function.

The structures revealed that Protocadherin-19 molecules from adjacent cells engage in a "forearm handshake" to form the bond that connects neurons. Some of the mutations that cause epilepsy occur in the region responsible for this Protocadherin-19 forearm handshake. Laboratory experiments confirmed that these mutations impair the formation of the adhesive bond, revealing the molecular basis for some of the mutations that underlie Protocadherin-19-female-limited epilepsy.

Other cadherin molecules may interact via a similar forearm handshake; this could be investigated in future experiments. It also remains to be discovered how brain wiring depends on Protocadherin-19 adhesion in animal development, and how altering these proteins can rewire developing brain circuits.

epilepsy patients, making it the second most clinically relevant gene in epilepsy. Approximately half of these mutations are missense mutations distributed throughout the extracellular domain of the PCDH19 protein. Despite the clear importance of *PCDH19* and other non-clustered δ-protocadherins to neural development, their specific roles are only beginning to be revealed. For example, Pcdh7, Pcdh17 and Pcdh18b are involved in axon outgrowth or arborization (*Piper et al., 2008*; *Hayashi et al., 2014*; *Biswas et al., 2014*), while several δ-protocadherins, including Pcdh19, regulate cell motility during early development (*Yamamoto et al., 1998*; *Aamar and Dawid, 2008*; *Biswas et al., 2010*; *Emond et al., 2009*). In zebrafish, *pcdh19*, regulates the formation of neuronal columns in the optic tectum, and loss of *pcdh19* degrades visually-guided behaviors (*Cooper et al., 2015*). However, it is not known how mutations in *PCDH19* lead to PCDH19-FE.

Cadherins typically mediate adhesion using their extracellular domains, which are made of two or more consecutive extracellular cadherin (EC) repeats (*Takeichi, 1990*; *Brasch et al., 2012*). The adhesion mechanism used by classical cadherins is well known and involves a tip-to-tip interaction that is stabilized by the reciprocal exchange of tryptophan residues at the N-terminal EC1 repeat most distant from the membrane (*Overduin et al., 1995*; *Shapiro et al., 1995*; *Nagar et al., 1996*; *Boggon et al., 2002*; *Patel et al., 2006*; *Zhang et al., 2009*; *Sivasankar et al., 2009*; *Harrison et al., 2010*; *Ciatto et al., 2010*; *Leckband and Sivasankar, 2012*). However, PCDH19 along with the rest of the non-classical cadherins lack these critical tryptophan residues and must mediate adhesion by an alternative mechanism (*Emond et al., 2011*; *Sotomayor et al., 2014*; *Biswas et al., 2010*). In the case of the non-classical protocadherin-15 and cadherin-23 proteins, an adhesive interface is formed by overlapping, antiparallel interactions of their EC1 and EC2 tips (*Elledge et al., 2010*; *Sotomayor et al., 2010*; *2012*; *Geng et al., 2013*). For clustered protocadherins, recent binding assays and structures suggest that adhesion is mediated by an antiparallel interaction of fully overlapping EC1 to EC4 domains (*Rubinstein et al., 2015*; *Nicoludis et al., 2015*; *Goodman et al., 2016*). Yet how non-clustered δ-protocadherins and PCDH19 form adhesive bonds and how these bonds are altered by disease-causing mutations is unknown.

Here we present crystals structures of the highly homologous zebrafish Protocadherin-19 (Pcdh19) encompassing repeats EC1-4 and EC3-4. The structures allow us to map >70% of the disease-causing missense mutations and provide a structural framework to interpret their functional impact. In addition, the structures suggest two possible homophilic adhesive interfaces, and complementary binding assays validate one of them, which is affected by multiple PCDH19-FE mutations. This interface involves fully overlapping EC1 to EC4 domains and likely represents a general interaction mechanism for the non-clustered δ-protocadherins.

## Results

To understand the mechanism of Pcdh19 function and to determine the structural role of PCDH19-FE mutations, the *Danio rerio* Pcdh19 EC1-4 and the EC3-4 fragments (70% identity, 83% similarity to *Homo sapiens* EC1-4) were produced in *E. coli,* refolded from inclusion bodies, and used for crystallization and structural determination (see Materials and methods). The solved structure for Pcdh19 EC3-4 (2.51 Å, *Table 1*, *Figure 1—figure supplement 1A*) includes four molecules in the asymmetric unit, each starting from Pro 213 and continuing to Asp 422 (numbering corresponds to the processed *Danio rerio* protein, see Materials and methods). Root-mean-square-deviation (RMSD) among these four molecules is <2.4 Å. One of the Pcdh19 EC3-4 molecules was used to solve the Pcdh19 EC1-4 structure (3.59 Å, *Table 1*, *Figure 1—figure supplement 1B*), which contains two molecules in the asymmetric unit, each starting from Val one to Asp 422 (RMSD of 1.4 Å). The EC3-4 repeats from both structures superpose well (RMSD 2.1 Å), and good quality electron density maps allowed us to unambiguously position side chains for most residues (Materials and methods and *Figure 1—figure supplement 1*). Given the similarities among our structures and chains, we will describe features as seen in the more complete chain B of Pcdh19 EC1-4, unless otherwise explicitly stated.

The architecture of all Pcdh19 EC repeats matches that observed for other cadherins (*Shapiro et al., 1995*; *Overduin et al., 1995*), with the typical Greek-key motif comprised of seven β strands (A-G) forming a β sandwich fold (*Figure 1A*). The EC1 repeat has a disulfide bond at the E-F loop, typical of clustered protocadherins, as well as one of two α-helices (at the B-C loop) also found in structures of clustered protocadherins (*Morishita et al., 2006*; *Nicoludis et al., 2015*; *Rubinstein et al., 2015*; *Goodman et al., 2016*) (*Figure 1A,B*). The three linker regions of Pcdh19 (EC1-2, EC2-3, EC3-4) have canonical cadherin calcium-binding sites (*Nagar et al., 1996*) (*Figure 1E–G*). Overall, our structures show canonical features and provide a unique framework to analyze >70% of the PCDH19-FE mutations.

### PCDH19-FE mutations analyzed in the context of the Pcdh19 EC1-4 structure

There are 51 PCDH19-FE missense mutations (out of 70) that can be mapped to 43 locations in the Pcdh19 EC1-4 structure (*Figure 1B* and *Figure 1—figure supplement 2*). These mutations can be classified in three groups. The first group (18 mutations at 14 locations) corresponds to residues whose side chains are pointing toward the hydrophobic core of an EC repeat (*Figure 1B,C*). The second group involves residues whose side chains are at the surface of the protein (10 mutations at 10 sites; *Figure 1B,D*). The last group includes residues at calcium-binding motifs, with 19 locations affected by 23 different mutations (*Figure 1B,E–G*). Mutations in each group are predicted to have different effects on the protein's structure (*Figure 1—figure supplement 3*).

PCDH19-FE mutations altering residues in the first group may often cause protein misfolding or structural instability. For instance, mutations L81R and I115K (corresponding to L58 and I92 in the crystal structure) would result in impossible conformations in which a positively charged residue side chain is pointing toward the hydrophobic core of EC1 (*Figure 1C*). Thus, these mutants are unlikely to fold properly. Mutation L25P (L4) will interfere with hydrogen bonding and secondary structure formation, while V72G (V50) is subtler, as it replaces a rather large hydrophobic residue with a different and smaller side chain that may only affect the packing of the EC1 hydrophobic core. The mutation A153T (A130) in EC2, in which a small hydrophobic residue is replaced by a larger hydrophilic threonine, may result in structural instability as well. A similar analysis can be done for all 18 mutations in this group (*Figure 1—figure supplement 3*). Protein misfolding and structural instability caused by these mutations are likely to inhibit PCDH19 adhesive function, either directly,

**Table 1.** Statistics for Protocadherin-19 structures.

| Data collection | DrPCDH19 EC1-4 | DrPCDH19 EC3-4 |
| --- | --- | --- |
| Space group | $P2_1$ | C2 |
| Unit cell parameters | | |
| a, b, c (Å) | 66.390, 59.776, 165.925 | 149.355, 86.631, 132.583 |
| $\alpha$, $\beta$, $\gamma$ (°) | 90, 94.39, 90 | 90, 122.13, 90 |
| Molecules per asymmetric unit | 2 | 4 |
| Beam source | MicroMax-003 | APS 24-ID-C |
| Date of data collection | 12-DEC-14 | 31-OCT-14 |
| Wavelength (Å) | 1.54187 | 0.97920 |
| Resolution (Å) | 50.00–3.59 (3.66–3.59) | 50.00–2.51 (2.55–2.51) |
| Unique reflections | 15416 | 47847 |
| Completeness (%) | 94.8 (86.0) | 98.0 (88.1) |
| Redundancy | 2.7 (2.4) | 4.4 (3.2) |
| $I / \sigma(I)$ | 4.90 (2.10) | 16.59 (2.21) |
| $R_{merge}$ | 0.182 (0.386) | 0.072 (0.591) |
| $R_{meas}$ | 0.224 (0.480) | 0.081 (0.690) |
| $R_{pim}$ | 0.129 (0.281) | 0.037 (0.348) |
| $CC_{1/2}$ | 0.833 (0.774) | 0.964 (0.792) |
| $CC^*$ | 0.966 (0.934) | 0.991 (0.940) |
| Refinement | | |
| Resolution range (Å) | 50.00–3.59 (3.68–3.59) | 50.00–2.51 (2.58–2.51) |
| $R_{work}$ (%) | 24.6 (41.3) | 18.8 (38.3) |
| $R_{free}$ (%) | 30.5 (45.2) | 23.9 (41.6) |
| Protein Residues | 839 | 827 |
| Ligands/ions | 20 | 16 |
| Water molecules | 15 | 49 |
| Rms deviations | | |
| Bond lengths (Å) | 0.0094 | 0.0112 |
| Bond angles (°) | 1.4661 | 1.3915 |
| *B*-factor average | | |
| Protein | 90.75 | 77.93 |
| Ligand/ion | 57.01 | 55.32 |
| Water | 45.40 | 60.92 |
| Ramachandran plot region (PROCHECK) | | |
| Most favored (%) | 78.6 | 85.6 |
| Additionally allowed (%) | 20.4 | 13.5 |
| Generously allowed (%) | 1.1 | 1.0 |
| Disallowed (%) | 0.0 | 0.0 |
| PDB ID code | 5IU9 | 5CO1 |

allosterically, or by altering the strength of cell-cell adhesion due to a reduced number of functional molecules on the cell surface.

The effect of ten PCDH19-FE mutations on residues with side chains at the protein surface (second group) is less clear. Two of them (S276P and L433P) may affect packing and folding, as these mutations to proline are predicted to prevent formation of hydrogen bonds important for β strand

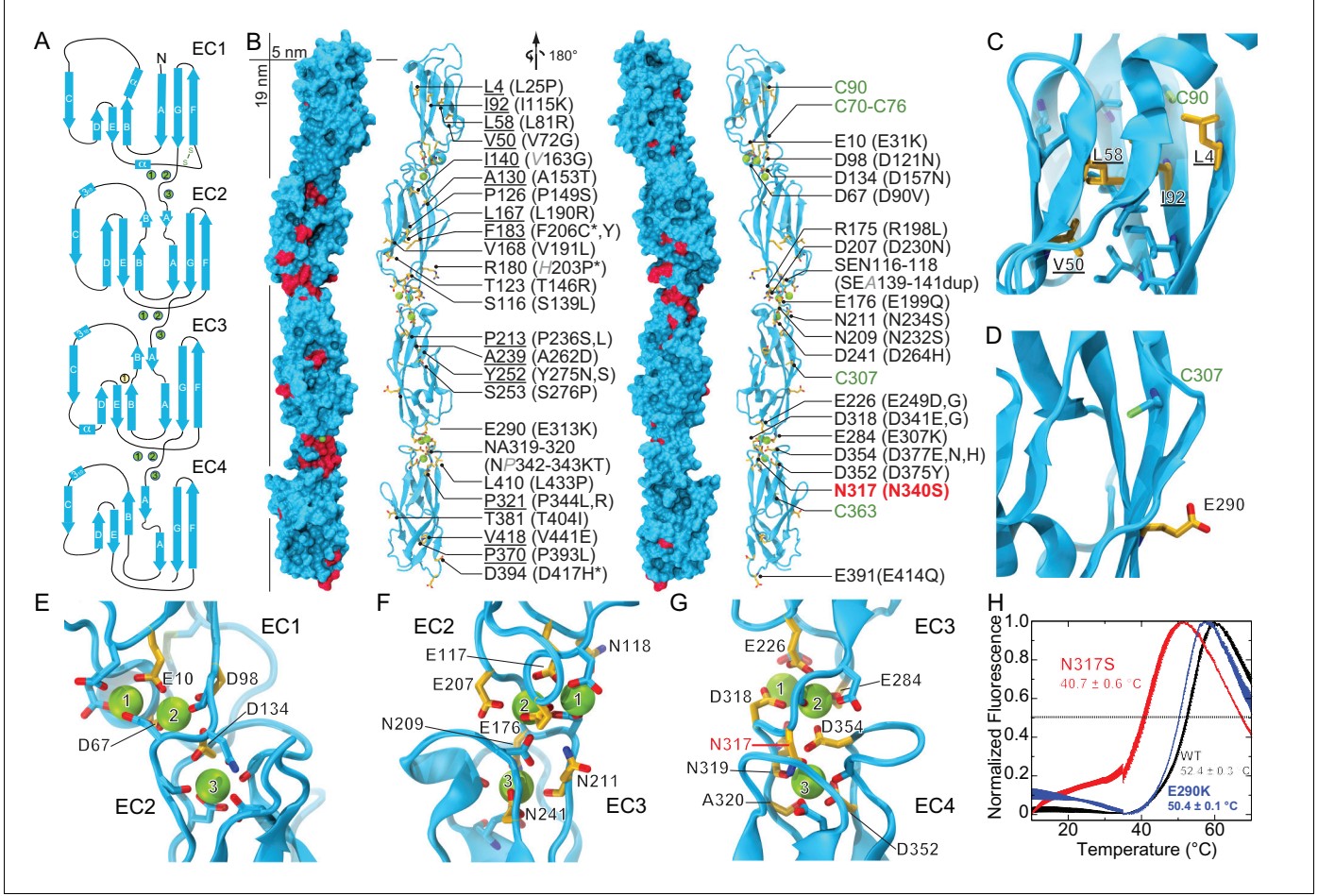

**Figure 1.** Pcdh19 EC1-4 structure reveals the location of PCDH19-FE missense mutations. (**A**) Topology diagram of Pcdh19 EC1-4. A typical cadherin fold is observed for each EC repeat with seven β strands labeled **A** to **G**. Calcium and sodium ions are shown as green and yellow circles, respectively. (**B**) Molecular surface representation and ribbon diagram of Pcdh19 EC1-4 shown in two orientations. Forty-three sites mutated in PCDH19-FE are highlighted in dark red on the protein surface (when applicable), shown in stick representation on the ribbon diagram, and listed. Mutations are indicated in parenthesis using the human gene numbering, with three non-conserved sites listed in *italic gray*. Residues whose side chains point to the protein core are underlined. Sites at inter-repeat, calcium-binding linker regions are listed on the right panel. The N317 site, involved in > 20 PCDH19-FE cases (N340S), is in red. Cysteine amino-acids are in lime; none are exposed. Paired mutations in single PCDH19-FE patients are indicated with a star (*). See also *Figure 1—source data 1*. (**C**) Detail of EC1 highlighting mutation sites (yellow sticks) in which residue side chains are pointing to the protein core. Neighboring hydrophobic core residues are shown in cyan. (**D**) Detail of EC3 highlighting a mutation site in which the residue side chain is exposed and pointing away from the protein surface. (**E-G**) Detail of calcium-binding inter-repeat linkers EC1-2 (**E**), EC2-3 (**F**), and EC3-4. (**G**) Calcium ions are shown in green and calcium-coordinating side chains in stick representation. Mutation sites are labeled and shown in yellow. (**H**) Melting temperature for the Pcdh19 EC3-4 wild type (WT) fragment, the N317S (equivalent to human N340S) and E290K (E313K) mutants determined using differential scanning fluorimetry. A clear decrease in thermostability is observed for the N317S mutant fragment in 2 mM CaCl₂, but not for the E290K mutant. The curves represent the average for each construct with vertical bars representing standard error of the mean. See also *Figure 1—figure supplement 1–3*.

The following source data and figure supplements are available for figure 1:

**Source data 1.** PCDH19-FE mutations.

**Figure supplement 1.** Electron density maps for the EC3-4 linker.

**Figure supplement 2.** Sequence alignment of zebrafish, mouse, and human Pcdh19 EC repeats.

**Figure supplement 3.** Predicted structural consequences of PCDH19-FE mutations.

formation and loop structure. Six of them are involved in putative homophilic interfaces, and their effect on binding is discussed below. The V191L mutation site is not directly involved in homophilic binding, but it is near residues that are, and may allosterically alter binding. Alternatively, this mutation may alter interactions with N-cadherin (*Emond et al., 2011*) or other PCDH19 molecular partners yet to be determined. The last mutation, D417H, is not involved in any known interface, but this epilepsy patient has a pair of mutations in PCDH19 (D417H and D596Y). It is unclear whether both mutations contribute to the epileptic syndrome (*Figure 1—source data 1*) (*Higurashi et al., 2015*; *Hoshina et al., 2015*).

The third group of mutations involves residues that are at one of the canonical calcium-binding motifs between EC repeats (XEX[BASE] and DRE from the first EC repeat, DXNDN from the linker, and DXD and XDX[TOP] from the second repeat). Two of these PCDH19-FE mutations involve charge reversal for a calcium-coordinating residue (E31K at X**E**X and E307K at D**Y**E), and may result in impaired folding and impaired calcium binding. Twelve PCDH19-FE mutations in this group replace a charged, calcium-coordinating residue by a neutral residue (D90V at DR**E,** D121N at **D**XNDN, D157N at DX**D**, E199Q at DR**E**, D230N at **D**XNDN, E249G at X**E**X, D264H at **D**XD, D341G at DXN**D**N, D375Y at **D**XD, D377N and D377H at DX**D**, and E414Q at DR**E**). Some of these mutations only affect charge, but not the size of the side chain (D to N and E to Q), and may decrease the affinity for calcium. Others involve more drastic side-chain size changes (D to Y or G) and will not only impair calcium binding, but might also induce protein instability. In addition, three mutations alter the size, but not the charge of a coordinating residue (E249D at X**E**X, D341E at DXN**D**N, and D377E DX**D**), indicating that even subtle perturbations at the calcium-binding linkers might result in impaired function. Three more PCDH19-FE mutations involve substituting a coordinating asparagine residue by a serine (N232S and N340S at DX**N**DN, and N234S at DXND**N**), with one of these mutations present in over twenty unrelated individuals (N340S). Similarly, the mutation NP342-343KT at DXND**N** involves a coordinating asparagine residue, but it is mutated to lysine and accompanied by a proline to threonine mutation. In addition, one mutation involves the non-calcium binding residue of the D**R**E motif (R198L), which may disrupt calcium binding. The last PCDH19-FE mutation in this group involves duplication of three residues (SEA139-141dup at X**E**X), one of which is directly coordinating calcium. This duplication might change the architecture of the loop and alter calcium binding as well. Overall, mutations at PCDH19 calcium-binding motifs are varied, with some predicted to have drastic effects on protein folding and calcium binding, and others predicted to have a minor effect yet still causing protein malfunction.

There are 19 PCDH19-FE missense mutations not found within EC1-4 (*Figure 1—figure supplement 2*), 14 of which are at conserved calcium-binding motifs (N557K, D594H, D596G,H,V,Y) or at other structurally conserved sites for cadherin repeats (P451L, G486R, G513R, L543P, P561R, G601D, V642M, L652P). Two mutations involve insertion or deletion of residues (N449_H450insN and S489del), and will likely disrupt β strand folding. However, the effect of the remaining three is unclear (R550P in β strand G of EC5, P567L in β strand A of EC6, and D618N likely at the end of β strand D); perhaps they are involved in *cis* interactions with PCDH19 or other cadherins.

To gain insights into the molecular mechanism of the most common PCDH19-FE mutation, N340S (N317S, *Figure 1—source data 1*), we introduced this mutation into the Pcdh19 EC3-4 construct and compared its thermal stability with the wild-type (WT) Pcdh19 fragment (*Figure 1H*). The Pcdh19 EC3-4 N317S fragment refolded well as assessed by size exclusion chromatography (SEC), but its melting temperature is considerably lower (40.7 ± 0.6°C *vs.* 52.4 ± 0.3°C), even in the presence of 2 mM CaCl$_2$. Another PCDH19-FE mutation of a surface residue (E313K, equivalent to E290K) did not show a dramatic shift in melting temperature (50.4 ± 0.1°C). These SEC and thermal stability results indicate that the EC3-4 fragment carrying the N317S mutation is folded, and may bind calcium, yet it is not as stable as the wild-type fragment.

## Antiparallel interfaces in crystal contacts of the Pcdh19 EC1-4 structure

Crystal structures have previously revealed the adhesive interfaces for classical cadherins, clustered protocadherins, and the protocadherin-15 and cadherin-23 complex (*Nagar et al., 1996*; *Boggon et al., 2002*; *Patel et al., 2006*; *Ciatto et al., 2010*; *Sotomayor et al., 2012*; *Nicoludis et al., 2015*; *Goodman et al., 2016*). Although the Pcdh19 EC3-4 structure does not reveal any relevant interface, the Pcdh19 EC1-4 structure does. The purified Pcdh19 EC1-4 fragment elutes in two well-defined peaks in size exclusion chromatography experiments (SEC), with these

peaks most likely representing monomeric and dimeric states in solution (*Figure 2—figure supplement 1*). Pcdh19 EC1-4 crystals were grown from the putative dimer SEC peak elution, and two plausible adhesive interfaces are observed in our Pcdh19 EC1-4 structure. The first one, which we will refer to as Pcdh19-I1, arises from contacts between the two Pcdh19 EC1-4 molecules in the asymmetric unit, and involves a fully-overlapped antiparallel dimer in which EC1 from one molecule interacts with EC4 from the other (EC1:EC4), EC2 with EC3 (EC2:EC3), EC3 with EC2 (EC3:EC2), and EC4 with EC1 (EC4:EC1; *Figure 2A,B*). Within the same protein crystal structure, the second antiparallel interface (Pcdh19-I2) involves the opposite side of Pcdh19 with observed EC2:EC4, EC3:EC3, and EC4:EC2 interactions, as well as potential (not observed) EC1:EC5 and EC5:EC1 contacts (*Figure 2—figure supplement 2A*). Several lines of evidence favor the first interface Pcdh19-I1 as the most likely to mediate biological function.

Analysis of the Pcdh19-I1 antiparallel interface with the Protein Interfaces, Surfaces and Assemblies (PISA) server (*Krissinel and Henrick, 2007*) and with the *NOXclass* classifier (*Zhu et al., 2006*) revealed a large interface (~1650 Å$^2$), that is unlikely to be a crystal packing artifact (89.21% biological, 81% obligate). In contrast, the possible antiparallel Pcdh19-I2 interface is predicted by *NOXclass* to be non physiological, as its smaller interface area (~930 Å$^2$) and the nature of its contacts matches those of crystal packing interactions (42.97% biological, 20.21% obligate). Yet, both interface areas are larger than 856 Å$^2$, an empirical cut-off that can distinguish biological interfaces from crystal contacts with 85% accuracy (*Ponstingl et al., 2000*), and our analysis of the Pcdh19-I2 interface lacks contributions from possible EC1-EC5 contacts, which might be significant. Moreover, shape correlation (*Lawrence and Colman, 1993*) for Pcdh19-I1 is lower than for the Pcdh19-I2 interface (S$_{c-I1}$ = 0.44 *vs.* S$_{c-I2}$ = 0.61), as there is a large gap between the main EC2-EC3:EC3-EC2 contacts and the EC1-EC4:EC4-EC1 interactions zones (*Figure 2A*).

To further differentiate between the possible Pcdh19-I1 and Pcdh19-I2 interfaces, we evaluated whether any of the six PCDH19-FE mutations altering surface residues at crystal contacts, but not necessarily protein structure, could interfere with binding. Five of these mutations (S139L, T146R, P149S, E313K, and T404I; all at conserved sites) change residues involved in the Pcdh19-I1 interface (S116, T123, P126, E290, T381 respectively in *Figure 2B–E*), where we define residues at a given interface as those with a buried surface area that is at least 20% of their accessible surface area according to PISA. In all cases we predict altered EC1-4 homophilic binding, as the size and nature of the residue is changed by each mutation (hydrophobic *vs.* hydrophilic; charged *vs.* non-charged). The remaining mutation (H203P) involves a non-conserved residue at the Pcdh19-I2 interface, which could impair its formation (R180 in *Figure 2—figure supplement 2A*). However, the patient with the H203P mutation also carries another PCDH19-FE mutation (F206C) at a location mutated in other epilepsy patients (*Marini et al., 2012*; *Depienne et al., 2011*); thus it is unclear if H203P is contributing to epilepsy. In contrast, all five PCDH19-FE mutations at the Pcdh19-I1 interface are likely causal, which suggests that Pcdh19-I1 is relevant *in vivo*.

We also analyzed predicted glycosylation sites that might interfere with binding and thereby reveal non-physiological interfaces, as observed for VE-cadherin (*Brasch et al., 2011*). There are 14 glycosylation sites within EC1-4, and none of them involve residues at the Pcdh19-I1 interface (*Figure 2—figure supplement 3A*). An O-linked glycosylation site is predicted to be at the Pcdh19-I2 interface (T232), and an additional O-linked glycosylation site is predicted for the human PCDH19 protein at S204 (the equivalent N202 in Pcdh19 is predicted to be non-glycosylated), also at the Pcdh19-I2 interface (*Figure 2—figure supplement 3B*). Glycation sites, for which sugar molecules might be added randomly and to long-lived proteins, are predicted at both interfaces (K156 and K308 in Pcdh19-I1 and K204 in Pcdh19-I2), but may not interfere directly with either, since glycation depends on environmental conditions and it has never been reported for cadherins (*Salahuddin et al., 2014*; *Simm et al., 2015*). Thus the lack of glycosylation sites at the Pcdh19-I1 interface renders it as the most likely to be functional.

While not conclusive, all the analyses presented above favor the Pcdh19-I1 antiparallel dimer over the Pcdh19-I2 interface in terms of physiological relevance. The larger surface area of the Pcdh19-I1 dimer, the nature of the residues involved, the number of PCDH19-FE mutations at this interface, and the lack of predicted glycosylation sites, all suggest that the Pcdh19-I1 interface may occur and be functional *in vivo*.

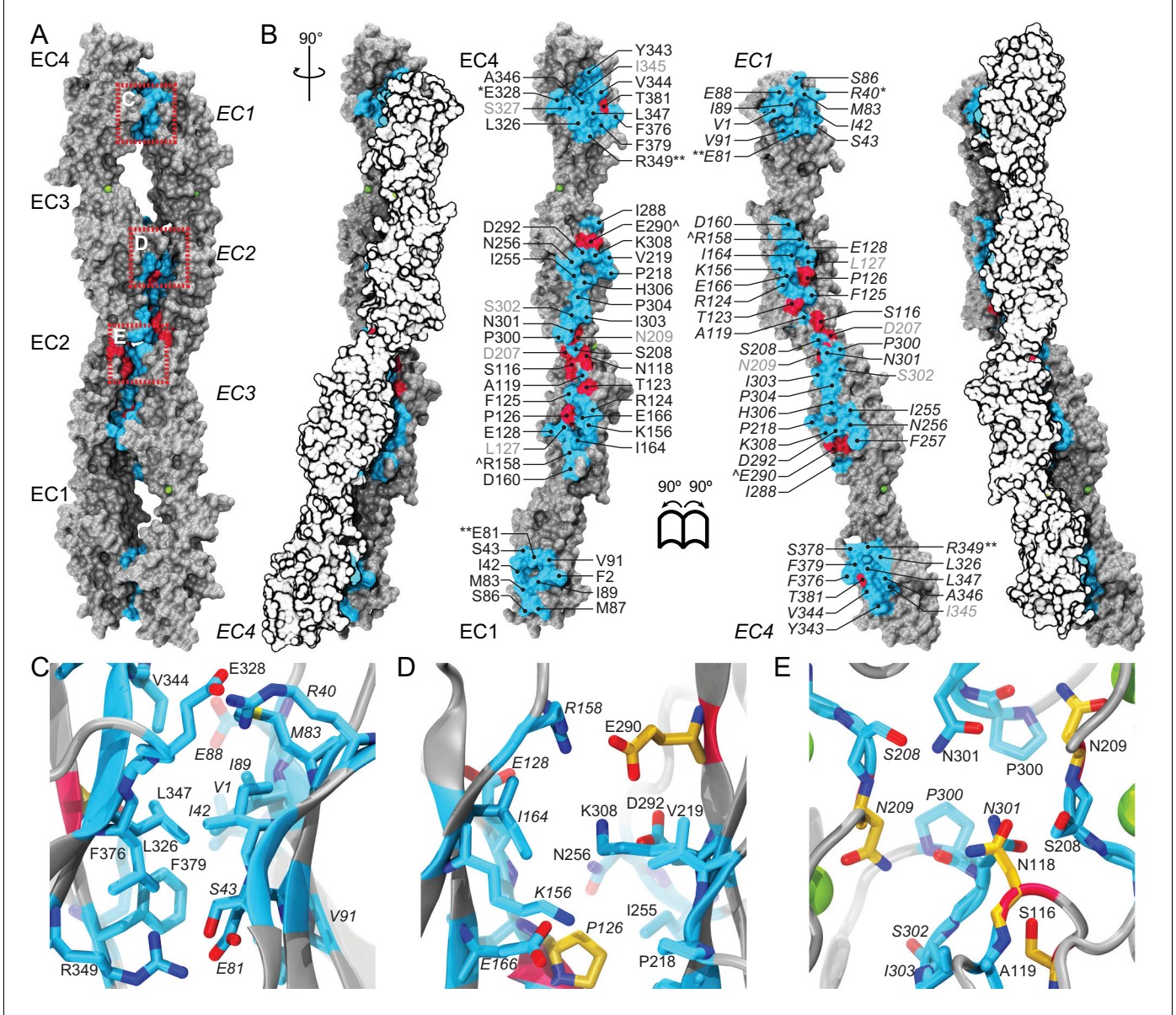

**Figure 2.** A crystallographic Pcdh19 antiparallel interface involves fully overlapped EC1-4 repeats. (**A**) Molecular surface representation of two Pcdh19 EC1-4 molecules interacting in the crystallographic asymmetric unit. In this dimeric arrangement, an interaction interface is formed by fully overlapped and antiparallel EC1-4 protomers (Pcdh19-I1). Red, dashed boxes indicate three interaction sites highlighted in panels (**C–E**). (**B**) Side views of the Pcdh19 dimer and the interaction surface exposed with interfacing residues listed and shown in cyan. Sites mutated in PCDH19-FE located at the interface are shown in dark red. Sites with residue side chains pointing to the protein core are labeled in gray text. Three inter-molecular salt bridges are indicated (*: R40-E328; **: E81-R349; ⌃: R158-E290). (**C–E**) Detail of antiparallel interface (red dashed boxes in **A**). Interfacing residues are in cyan and yellow (PCDH-FE). Left panel is in the same orientation as **A**, middle and right panels are rotated around the dimer's longest axis. Labels for one of the protomers are in *italics*. See also **Figure 2—figure supplement 1–3**.

The following figure supplements are available for figure 2:

**Figure supplement 1.** Two states for Pcdh19 EC1-4 in solution.

**Figure supplement 2.** Alternate crystallographic antiparallel interface involves EC1 to EC5 repeats.

**Figure supplement 3.** Pcdh19 dimer interfaces and predicted glycosylation and glycation sites.

## Binding assays probing Pcdh19 interfaces

To conclusively test which binding interface mediates Pcdh19 adhesion, and whether PCDH19-FE mutations at the protein surface can interfere with one of the two possible Pcdh19 interfaces described above, we used modified bead aggregations assays, mutagenesis, and size exclusion chromatography experiments. Previous cell-based assays showed weak homophilic adhesion for the chicken Pcdh19 (*Tai et al., 2010*). In addition, previous assays in which the full-length Pcdh19 extracellular cadherin domain fused to Fc (Pcdh19ECFc) was incubated with protein A beads showed calcium-dependent aggregation only when it was co-purified with N-cadherin (*Biswas et al., 2010*; *Emond et al., 2011*). To study Pcdh19 homophilic interactions, we modified the previous protocol (*Emond and Jontes, 2014*) and added a final step in which beads were rocked (*Sano et al., 1993*) in a controlled fashion for up to two minutes (see Materials and methods). The modified protocol allowed us to identify clear bead aggregates mediated by Pcdh19ECFc alone (*Figure 3—figure supplement 1*).

To identify the minimal adhesive unit of Pcdh19 we used our modified protocol with truncated versions of Pcdh19 containing different numbers of EC repeats: Pcdh19ECFc (EC1-6), Pcdh19EC1-5Fc, Pcdh19EC1-4Fc, Pcdh19EC1-3Fc, Pcdh19EC1-2Fc, and Pcdh19EC2-6Fc (*Figure 3*). Bead aggregation was observed only when using Pcdh19ECFc, Pcdh19EC1-5Fc, and Pcdh19EC1-4Fc, thus suggesting that Pcdh19EC1-4 is the minimal adhesive unit and highlighting the biological relevance of the antiparallel Pcdh19-I1 interface, which involves EC1-4 only.

Next, we introduced two PCDH19-FE mutations (T146R and E313K located at the Pcdh19-I1 interface; T123R and E290K in *Figure 2B*) in the full-length Pcdh19 extracellular domain and tested bead aggregation with these protein constructs (*Figure 4*). Bead aggregates were not detected

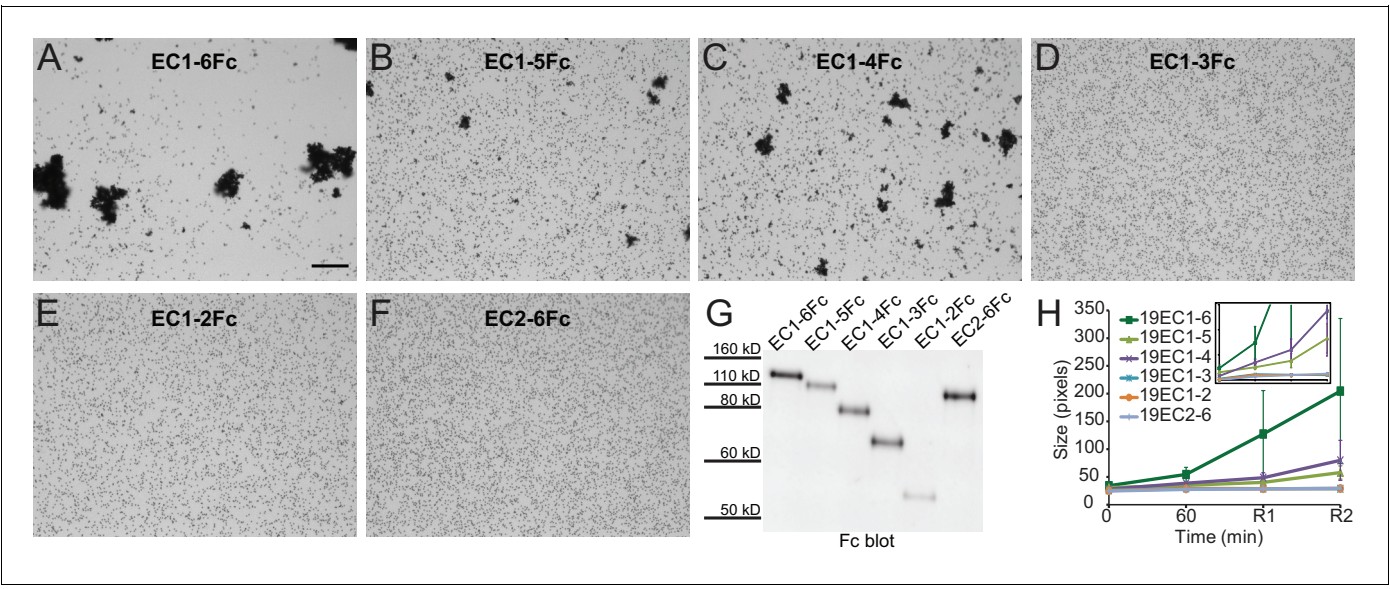

**Figure 3.** Minimal adhesive Pcdh19 fragment includes repeats EC1-4. (**A–F**) Protein G beads coated with full-length (**A**) and truncated versions (**B–F**) of the Pcdh19 extracellular domain imaged after incubation for 1 hr followed by rocking for 2 min in the presence of calcium. Bar – 100 μm. (**G**) Western blot shows efficient production and secretion of full-length and truncated Pcdh19 extracellular domains. (**H**) Aggregate size for full-length and truncated versions of the Pcdh19 extracellular domain after 1 hr of incubation followed by rocking for 1 min (R1) and for 2 min (R2). Error bars are standard error of the mean (n = 3 for all aggregation assays and constructs). Inset: zoom-in showing pixel size from 15 to 85 (*y* axis). Bead aggregation was observed for constructs including EC1-6Fc, EC1-5Fc, and EC1-4Fc. Data for EC1-6 is also plotted in *Figure 2—figure supplement 2D* (WT), *Figure 3—figure supplement 1C* (Pcdh19ECFc (Ca²⁺)), and *Figure 4H* (WT (Ca²⁺)) for comparison to additional constructs. See also *Figure 3—source data 1*.

The following source data and figure supplement are available for figure 3:

**Source data 1.** Quantification of aggregation assays.

**Figure supplement 1.** Modified bead aggregation assays can detect calcium-dependent homophilic Pcdh19 interactions.

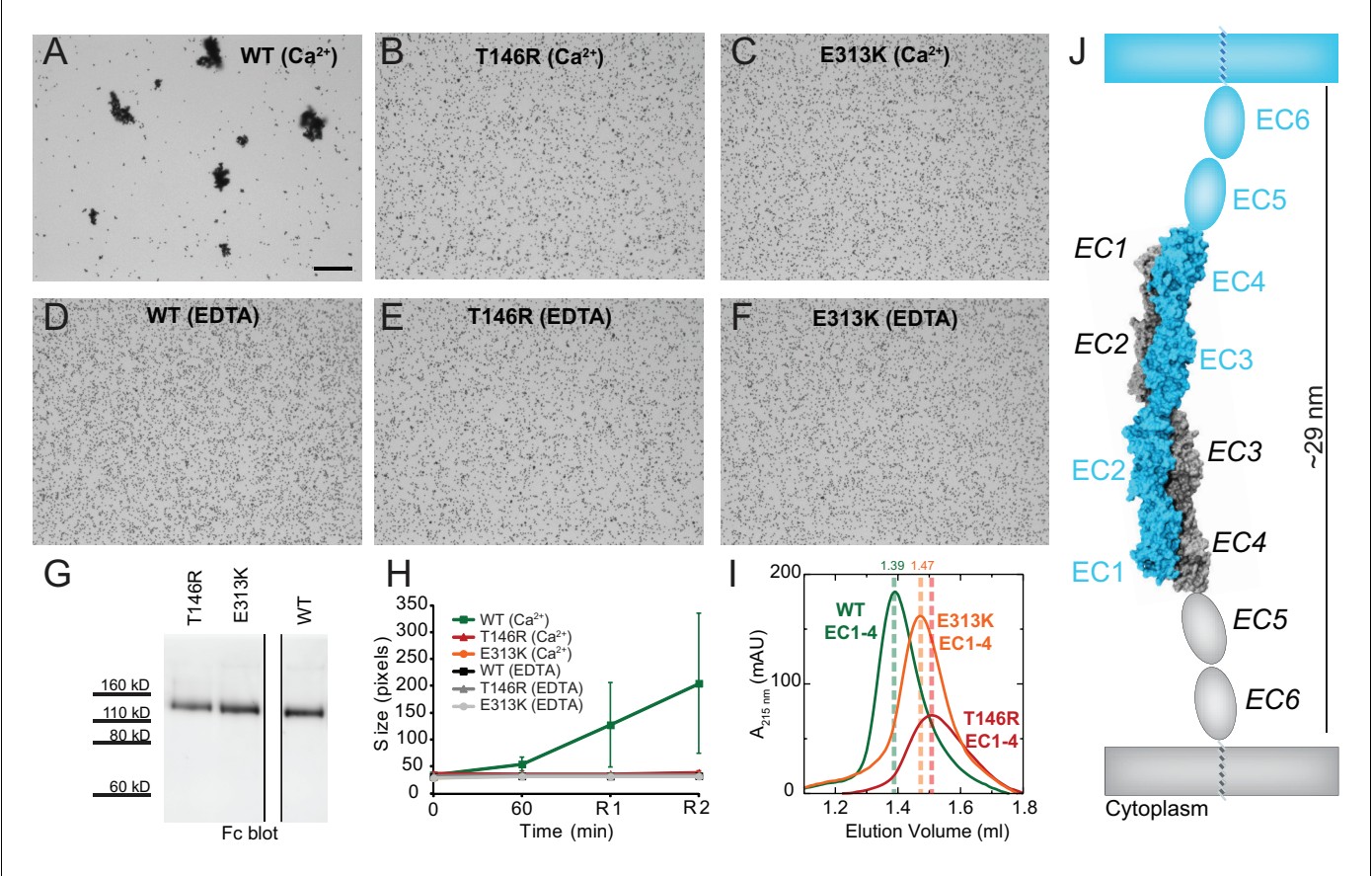

**Figure 4.** PCDH19-FE mutations at Pcdh19 I1 antiparallel interface impair Pcdh19-mediated bead aggregation. (**A–F**) Protein G beads coated with full-length extracellular wild-type (WT) Pcdh19ECFc (**A**) and two PCDH19-FE mutants (**B,C**) imaged after incubation for 1 hr followed by rocking for 2 min in the presence of calcium. Representative images for parallel experiments in the absence of calcium are shown in panels D to F (EDTA). All full-length extracellular domains were produced in HEK293 cells. Bar – 100 μm. (**G**) Western blot shows efficient production and secretion of both WT and mutant proteins used for bead aggregation assays. Parallel black lines indicate a two-lane gap between samples. (**H**) Aggregate size for WT and PCDH-FE mutants in the presence ($Ca^{2+}$) and absence (EDTA) of calcium after 1 hr of incubation followed by rocking for 1 min (R1) and for 2 min (R2). Error bars are standard error of the mean (n = 3 for all aggregation assays and constructs, *Figure 3—source data 1*). Aggregation is only observed for Pcdh19 WT in the presence of calcium and after rocking (see also *Figure 3H*). (**I**) Analytical size exclusion chromatogram traces for WT (green) and mutant (orange and red) Pcdh19 EC1-4 fragments produced in *E. coli*. A shift in peak elution volumes indicates impaired homophilic interaction for mutants. (**J**) Schematics of proposed homophilic 'forearm handshake' for the Pcdh19 adhesion complex validated through binding assays with protein carrying PCDH19-FE mutations. See also *Figure 4—figure supplement 1–2*.

The following figure supplements are available for figure 4:

**Figure supplement 1.** PCDH19-FE mutations at Pcdh19-I1 impair bead aggregation even in the presence of N-cadherin.

**Figure supplement 2.** PCDH19-FE mutations at Pcdh19-I1 do not abolish the interaction between the extracellular domains of Pcdh19 and N-cadherin.

when the Pcdh19ECFc carried these mutations under the conditions tested (*Figure 4B–C,E–H*). In contrast, the mutation R364E, predicted to impair the Pcdh19-I2 interface, did not eliminate bead aggregation (*Figure 2—figure supplement 2B–D*). Moreover, the presence of a N-cadherin (Ncad) fragment known to enhance Pcdh19-mediated adhesion (*Emond et al., 2011*) did not qualitatively change the effect of the T146R and E313K mutations. Bead aggregates were greatly diminished for T146R and abolished for E313K in the presence of NcadEC W2A/R14E His, a non-adhesive Ncad mutant previously used to study Pcdh19-mediated homophilic adhesion (*Harrison et al., 2010*; *Emond et al., 2011*) (*Figure 4—figure supplement 1B–C,E–F*). In addition, these mutations did not abolish the interaction between Pcdh19ECFc and NcadEC W2A/R14E His (*Figure 4—figure*

*supplement 2*). It is possible that the T146R and 313K mutations affect interactions with N-cadherin in a subtle way (directly or allosterically), yet our experimental results suggest that these mutations directly impair Pcdh19 homophilic adhesion.

We also introduced the T146R and E313K mutations at the Pcdh19-I1 interface into the bacterially expressed Pcdh19 EC1-4 protein fragment, and used analytical size exclusion chromatography to determine whether the mutant fragments were eluting as putative dimers or monomers in solution. Both mutations resulted in a shift of the elution peak that indicated a smaller, monomeric state (*Figure 4I*). Taken together, our crystallographic structural analyses and binding assays including PCDH19-FE mutations strongly support a model in which fully overlapped EC1-4 domains (Pcdh19-I1 interface) form the functional adhesive unit of Pcdh19 (*Figure 4J*).

## Model for PCDH19 adhesive interaction and implications for other protocadherins

The antiparallel Pcdh19-I1 dimer interface validated above reveals a homophilic 'forearm handshake' binding mechanism for PCDH19, involving overlap of 4 ECs from each protomer wrapping around each other. This is different from the mechanism used by classical cadherins, only involving EC1 (*Brasch et al., 2012*) or the heterophilic 'extended handshake' used by protocadherin-15 and cadherin-23, involving overlap of only EC1-2 of each protein (*Sotomayor et al., 2012*). The forearm handshake is similar to the binding mechanism recently reported for clustered protocadherins (*Goodman et al., 2016*; *Rubinstein et al., 2015*; *Nicoludis et al., 2015*) and might be used by other non-clustered protocadherins.

The Pcdh19-I1 interface involves extended and mostly symmetric, in-register contacts between repeats EC2:EC3 that account for ~58% of the interfacial area, as well as smaller, separate EC1:EC4 contacts (~350 Å$^2$ × 2) that are slightly off-register. The EC1:EC4 contacts arise as both repeats bend to meet after the C-terminal end of EC3 and the N-terminal end of EC2 separate from each other. In this arrangement, the EC2-3 linkers from each protomer are right next to each other, while the EC3-EC4 linker in one protomer is separated from the EC1-2 linker of the binding partner by a large opening. The interface is generally amphiphilic, with ~49% of the interfacial area involving hydrophobic residues, ~28% hydrophilic, and ~23% charged residues (*Figure 5—figure supplement 1*). Interestingly, the contact formed by EC1:EC4 is more hydrophobic (58%; 22%; 20%) than the one formed by EC2:EC3 (41%; 33%; 26%), yet salt-bridge pairs across protomers are present in both: R40-E328 and E81-R39 enhance the EC1 to EC4 contacts (*Figure 2C*) and R158-E290 links EC2 to EC3 (*Figure 2D*). While the R40-E328 pair seems to be zebrafish specific, the other two salt-bridges are highly conserved across sequenced species, along with most of the residues involved in the Pcdh19 EC1-4 interface (*Figure 5A* and *Figure 5—figure supplement 2*). The same set of residues is highly variable across different members of the δ1, δ2, and the clustered protocadherins (*Figure 5B* and *Figure 5—figure supplement 3*), suggesting that binding mechanisms might differ across subfamilies or that residue variability might confer specificity within a common binding mechanism.

A comparison of our Pcdh19-I1 interface to recently reported models and structures of clustered protocadherin interfaces (*Nicoludis et al., 2015*; *Goodman et al., 2016*) reveals multiple similarities among them. The most complete models of α and β-protocadherins show similar, fully overlapped antiparallel EC1-4 dimers (*Figure 5—figure supplement 4A–D*), with the same extended EC2:EC3 antiparallel connection accompanied with smaller EC1:EC4 contacts and salt-bridges across protomers. Structural alignments show that the relative arrangements of protomers within the antiparallel dimers for Pcdh19, *Mm* Pcdhα4 (5DZW), and *Mm* Pcdhα7 (5DZV) are the most similar to each other with slight shifting in some EC repeats (*Figure 5—figure supplement 4A,B*). The *Mm* Pcdhβ6 (5DZX) and *Mm* Pcdhβ8 (5DZY) structures show similar dimeric interfaces, but the relative arrangement of protomers within the dimer is slightly shifted for all EC repeats (*Figure 5—figure supplement 4C,D*). Similarly, the *Mm* PcdhγA1 EC1-3 interface (4ZI9) matches and aligns well with the Pcdh19 EC1-4 dimer (*Figure 5—figure supplement 4E*). Mapping of all interaction sites to the Pcdh19 EC1-4 topology diagram (*Figure 5C*) reveals a pattern for common interacting domains in odd and even EC repeats across these structures, which include the F-G β hairpin and β strand A for repeats EC1 and EC3, as well as the A-B and D-E β hairpins for EC2 and EC4. While there are differences in some of the interacting domains, dimeric arrangements, and contact details, including

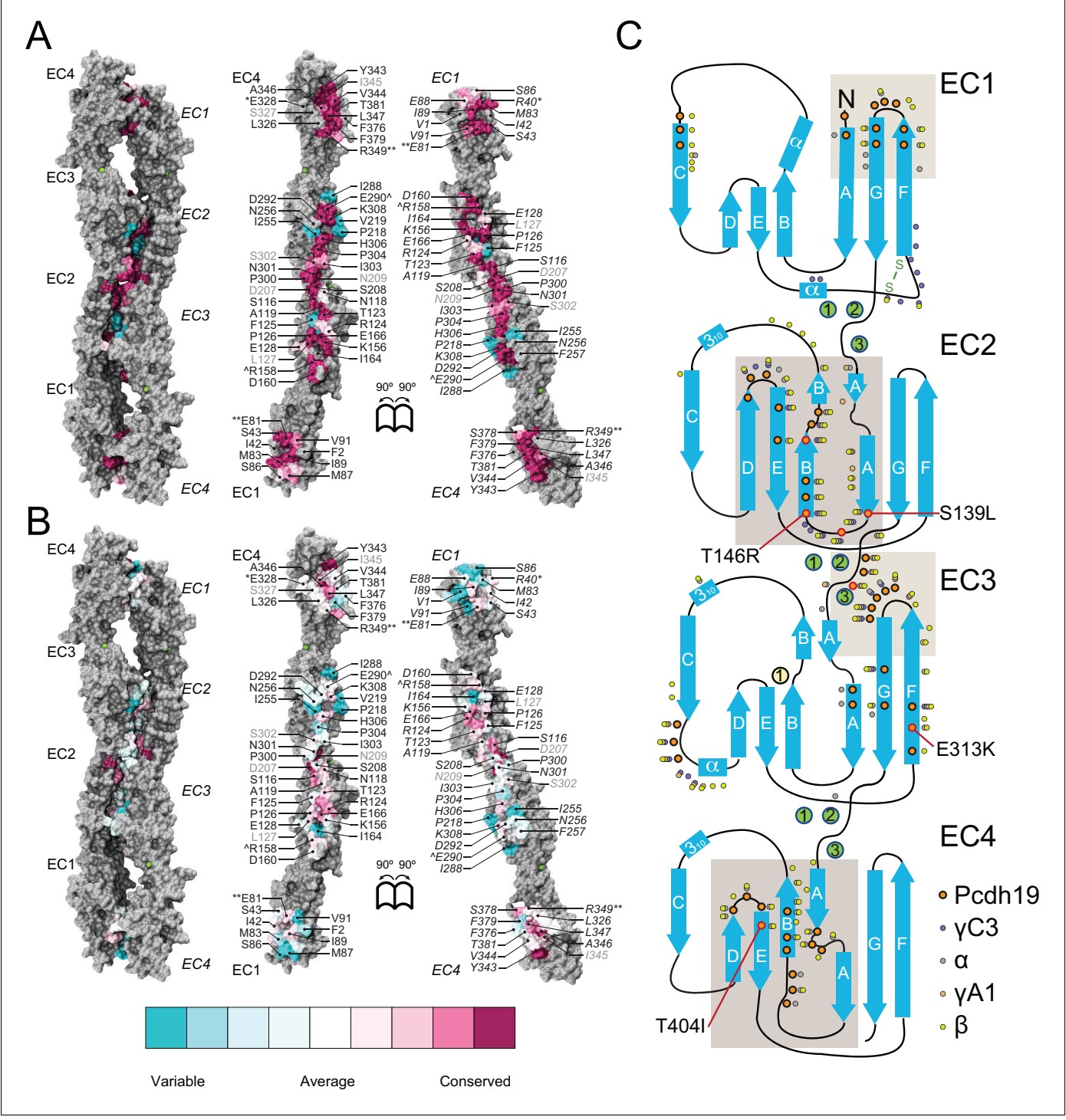

**Figure 5.** A common binding mechanism with sequence-diverse interfaces for δ and clustered protocadherins. (**A**) Molecular surface representation of the closed (left) and exposed (right) Pcdh19-I1 antiparallel dimer. Interfacing residues are colored according to sequence conservation among 102 species (*Figure 5—figure supplement 2* and *Figure 5—source data 1*). Most of them are highly conserved. Labels as in *Figure 2B*. (**B**) Antiparallel Pcdh19 EC1-4 dimer shown as in (**A**), with interfacing residues colored by sequence conservation among selected members of the non-clustered δ1- and δ2-protocadherins, as well as selected α, β, and γ clustered protocadherins (*Figure 5—figure supplement 3* and *Figure 5—source data 2*). (**C**) Location of interfacing residues for Pcdh19, *Mm* pcdhγC3, *Mm* pcdhα4 and α7, *Mm* pcdhγA1, and *Mm* pcdhβ6 and β8, mapped onto the Pcdh19 topology diagram. Shared structural motifs involved in binding include: The F-G loop along with the beginning of β strands A, G and C in EC1; the A-B loop, most of β strand B, the D-E loop, and the beginning of β strand E in EC2; the EC2-3 linker; the C-D loop, parts of β strands F and G and the F-G

*Figure 5 continued on next page*

*Figure 5 continued*

loop in EC3; the loop within β strand A, β strand B, and the D-E loop in EC4. Red/orange circles indicate sites mutated in PCDH19-FE. Common contact zones in EC1 and EC3, as well as EC2 and EC4, are highlighted with a brown background. See also *Figure 5—figure supplement 1–5*.

The following source data and figure supplements are available for figure 5:

**Source data 1.** Protocadherin-19 sequences.
**Source data 2.** Sequences for selected clustered and δ-protocadherins.
**Figure supplement 1.** Pcdh19-I1 antiparallel EC1-4 dimer interface involves charged, hydrophilic, and hydrophobic residues.
**Figure supplement 2.** Sequence alignment of Pcdh19 EC1-4.
**Figure supplement 3.** Sequence alignment of selected protocadherins.
**Figure supplement 4.** Structural comparison of Pcdh19-I1 EC1-4 dimer to clustered-protocadherin dimers.
**Figure supplement 5.** Structural comparison of protocadherin δ1 and δ2 EC3 repeats.

diversity of interfacial residues, clustered protocadherins seem to use the same binding mechanism that Pcdh19 uses to mediate adhesion.

A conserved RGD sequence motif within Pcdh19 EC2 (residues 158 to 160) at its D-E loop is similar to an integrin-binding RGD site within EC1 (C-D loop) in the α-protocadherins (*Ruoslahti, 1996*; *Mutoh et al., 2004*). The EC1 RGD motif is exposed in the *Mm* Pcdhα4 and *Mm* Pcdhα7 homodimers while the EC2 RGD motif of Pcdh19 (also present in Pcdh17 [*Kim et al., 2011*]) is buried at the EC2:EC3 contacts in the Pcdh19-I1 interface. This suggests that homophilic binding could regulate the availability of this potential, untested, integrin-binding site.

Pcdh19 belongs to the δ2-protocadherin subfamily, and given the sequence similarity among subfamily members, it is likely that all use the same dimer interface to mediate adhesion. This is less obvious for the δ1 subfamily, with members that have seven EC repeats and that display some critical differences at interaction sites, such as the presence of a positively charged residue (R or K) at position 290, where most δ2 members have a negatively charged glutamate that interacts with an arginine at position 158 (*Figure 2D*, *Figure 5—figure supplements 3* and *5*). The PCDH19-FE E313K mutation at this site (E290) prevents binding (*Figure 4C,H–I* and *Figure 4—figure supplement 1*), suggesting that δ1-protocadherins, which effectively carry the same mutation, should use a different interface to mediate adhesion. Yet, residues at position 157 and 158 in δ1-protocadherins are also charge swapped, with aspartates and glutamates that would restore this critical salt-bridge interaction at the EC2:EC3 interface, and at the same time prevent heterophilic interactions with δ2-protocadherins (*Figure 5—figure supplements 3* and *5*). Thus, it is likely that all non-clustered δ-protocadherins use fully overlapped EC1-4 antiparallel interfaces, like the one observed for Pcdh19, to mediate adhesion.

## Discussion and conclusions

The non-clustered δ-protocadherins are increasingly linked to human neurodevelopmental disorders, emphasizing both their importance to brain development and their relevance to human health (*Redies et al., 2005*; *Redies et al., 2012*; *Hirabayashi and Yagi, 2014*). In particular, mutations in *PCDH19* cause a female-limited form of infant-onset epilepsy (*Dibbens et al., 2008*; *Scheffer et al., 2008*; *Depienne and LeGuern, 2012*; *van Harssel et al., 2013*; *Leonardi et al., 2014*; *Thiffault et al., 2016*; *Terracciano et al., 2016*). Therefore, it is imperative to understand the developmental roles of PCDH19 and other non-clustered δ-protocadherins, the structural basis of homophilic adhesion by these molecules, and the functional impact of pathogenic missense mutations. The structural and biochemical data presented here provide a first view on the molecular mechanism of Pcdh19 adhesion, which is likely used by all non-clustered δ and clustered protocadherins.

Moreover, our Pcdh19 EC1-4 structural model shows > 70% of the missense mutations identified in PCDH19-FE patients, and reveals the biochemical basis for the deleterious effects for many of these mutations.

The Pcdh19 EC1-4 structure reveals an antiparallel dimer that is consistent with a *trans* adhesive interface, a conclusion supported by multiple lines of evidence. Notably, several missense mutations identified in PCDH19-FE patients localize to this interface. Two of these missense mutations (T146R and E313K) impair dimerization, as assessed by analytical gel filtration, and adhesion as assessed in bead aggregation assays, with and without N-cadherin. Sequence analysis suggests that the antiparallel adhesive mechanism presented here is broadly relevant to other, related δ-protocadherins. Recent work with clustered protocadherins, implicated in self-avoidance and self/non-self recognition (*Lefebvre et al., 2012*; *Kostadinov and Sanes, 2015*; *Yagi, 2012*), have revealed a similar antiparallel adhesive interface for these clustered protocadherins (*Rubinstein et al., 2015*; *Nicoludis et al., 2015*; *Goodman et al., 2016*). Thus, the Pcdh19-I1 adhesive interface observed in our Pcdh19 EC1-4 structure likely represents the mechanism used by both non-clustered δ-protocadherins and clustered protocadherins, which, together, represent the largest group within the cadherin superfamily.

Our structural data for Pcdh19, as well as recent work with the clustered protocadherins raises an interesting conundrum. The adhesive interface for protocadherins is extensive and involves interactions extending throughout EC1-4. This contrasts sharply with the adhesive interface of classical cadherins, which is restricted to EC1 and involves the reciprocal swap of Aβ-strands that is stabilized by burying Trp2 in a hydrophobic pocket (*Brasch et al., 2012*). However, the $K_D$ for dimerization of α- and β-protocadherins is in the micromolar range (similar to classical cadherins), bead aggregation and cell-based assays have consistently shown weak adhesion by both non-clustered and clustered protocadherins, and protocadherins are widely recognized as being only weakly adhesive (*Schreiner and Weiner, 2010*; *Thu et al., 2014*; *Sano et al., 1993*; *Rubinstein et al., 2015*). This disparity suggests that other mechanisms could modulate protocadherin adhesion *in vivo*. For instance, *cis*-oligomerization could compete with *trans* adhesive interactions, or interactions with other proteins, including N-cadherin, could sequester protocadherins or mask their adhesive interface. Further studies will be required to better understand protocadherin adhesion, how it may be altered in the presence of N-cadherin, and how it is regulated *in vivo*.

In addition to mutations that disrupt adhesion, our data reveal the potential effects of two other classes of mutations. In the first class, many mutations are predicted to directly impair folding and stability, which could lead to reduced levels of protein on the surface, due to impaired trafficking or enhanced protein degradation. In the second, PCDH19-FE mutations affecting calcium-binding sites are likely to cause shifts in calcium affinity as well as protein instability. Similar mutations in cadherin-23 and protocadherin-15 have been shown to decrease protein affinity for calcium, with $K_D$ shifts that are relevant in the context of the low calcium concentration to which these proteins are exposed (*Sotomayor et al., 2010*). Yet PCDH19 is expected to be in interstitial space with high calcium concentration, so it is more likely that the relevant effect of PCDH19-FE mutations at calcium-binding sites is compromised stability (even at saturating calcium concentrations), as shown here for the N340S mutation. Finally, analysis of one PCDH19-FE mutation within EC1-4, and three within EC5-6, reveal no obvious predicted consequences at the structural level, as they are exposed residues that should not affect calcium-binding, protein stability or adhesion. These mutations may impact a variety of protein-protein interactions. Although the physiological relevance is unclear, both non-clustered and clustered protocadherins can form *cis*-homo- or *cis*-hetero-oligomers (*Chen et al., 2007*; *Schreiner and Weiner, 2010*), and mutations affecting the formation of *cis*-oligomers could adversely impact protocadherin function. Similarly, protocadherins participate in a variety of protein complexes beyond homophilic *trans* adhesion: Pcdh19 has been shown to associate in *cis* with N-cadherin (*Emond et al., 2011*); protocadherins associate with the Wnt co-receptor, RYK (*Berndt et al., 2011*); PAPC interacts with Frizzled-7 and FLRT3 (*Chen et al., 2009*; *Kraft et al., 2012*); and Pcdh17 and Pcdh19 have highly conserved RGD sequences, suggesting that they may interact with integrins (*Ruoslahti, 1996*; *Mutoh et al., 2004*; *Kim et al., 2011*). Thus, further experimental characterization of key mutants *in vitro* and *in vivo* will continue to reveal correlations between structural defects, cellular-level defects, and different aspects of PCDH19-FE.

The non-clustered protocadherins are increasingly recognized as a family of molecules that play important roles during neural development. In addition to the role of PCDH19 in epilepsy, mutation

of *PCDH12* was found to underlie a syndrome of microcephaly that is associated with epilepsy and developmental disability (*Aran et al., 2016*). Moreover, both *PCDH9* and *PCDH10* have been associated with autism spectrum disorders (*Marshall et al., 2008*; *Morrow et al., 2008*). Ongoing work will likely uncover further links between members of this family and neurodevelopmental disorders. Our Pcdh19 EC1-4 model is the first to show the structural basis of adhesion by the non-clustered δ-protocadherins, and reveals that some of the missense mutations identified in PCDH19-FE occur at the adhesive interface and act by abolishing adhesion. This represents an initial stage in understanding the mechanisms of non-clustered δ-protocadherin homophilic adhesion and provides insight into the biochemical basis of protocadherin-based neurodevelopmental disease.

## Materials and methods

### Cloning and mutagenesis

Zebrafish Pcdh19 repeats EC1-4 and EC3-4 were subcloned into NdeI and XhoI sites of the pET21a vector for bacterial expression. Constructs for mammalian expression were created from previously reported constructs (Pcdh19, Pcdh19EC, Ncad, and NcadECW2A/R14E) and cloned into CMV:N1-Fc and CMV:N1-His backbones, respectively (*Biswas et al., 2010*; *Emond et al., 2011*). Truncated versions of Pcdh19 (Pcdh19EC1-5, Pcdh19EC1-4, Pcdh19EC1-3, Pcdh19EC1-2, Pcdh19EC2-6) were created by PCR subcloning of a Kozak sequence (GCCACC), the signal peptide, and appropriate EC domains into CMV:N1-Fc. Mutations were created in both the bacterial and mammalian expression constructs by site-directed mutagenesis. All constructs were sequence verified.

### Expression and purification of Pcdh19 fragments for structural determination

Each construct was expressed in BL21CodonPlus(DE3)-RIPL cells (Stratagene), cultured in TB (EC1-4) or LB (EC3-4), induced at $OD_{600}$ = 0.6 with 100 µM (EC1-4) or 200 µM (EC3-4) IPTG and grown at 30°C (EC1-4) or 25°C (EC3-4) for ~16 hr. Cells were lysed by sonication in denaturing buffer (20 mM TrisHCl [pH7.5], 6 M guanidine hydrochloride, 10 mM $CaCl_2$ and 20 mM imidazole). The cleared lysates were loaded onto Ni-Sepharose (GE Healthcare, Sweden), and eluted with denaturing buffer supplemented with 500 mM imidazole. Pcdh19 EC3-4 was refolded by overnight dialysis against 20 mM TrisHCl [pH 7.5], 150 mM NaCl, 400 mM arginine, 2 mM $CaCl_2$, 2 mM DTT using MWCO 2000 membranes. Pcdh19 EC1-4 was refolded by iterative dilution of the denaturing buffer with refolding buffer (100 mM TrisHCl [pH 8.5], 10 mM $CaCl_2$) (*Dechavanne et al., 2011*). Refolded protein was further purified on a Superdex200 column (GE Healthcare) in 20 mM TrisHCl [pH 8.0], 150 mM NaCl, 2 mM $CaCl_2$ and 1 mM DTT.

### Crystallization, data collection and structure determination

Crystals were grown by vapor diffusion at 4°C by mixing equal volumes of protein (Pcdh19 EC3-4 = 14.4 mg/ml and Pcdh19 EC1-4 = 7.7 mg/ml) and reservoir solution (Pcdh19 EC3-4 contained 100 mM calcium acetate, 100 mM sodium cacodylate [pH 6.1], 25% MPD; Pcdh19 EC1-4 contained 200 mM sodium chloride, 100 mM TrisHCl [pH 8.1], 8% PEG 20,000). Crystals were cryoprotected in reservoir solution (Pcdh19 EC3-4) or with 25% glycerol added (Pcdh19 EC1-4), and then cryo-cooled in liquid $N_2$. X-ray diffraction data were collected as indicated in *Table 1* and processed with HKL2000 or HKL3000 (*Minor et al., 2006*). The Pcdh19 EC3-4 structure was determined by molecular replacement using separate homology models for each repeat (4AQE_A for EC3 and 1L3W for EC4) as an initial search model using MrBUMP (*Keegan and Winn, 2007*) and PHASER (*McCoy et al., 2007*). Model building was done with COOT (*Emsley et al., 2010*) and restrained TLS refinement was performed with REFMAC5 (*Murshudov et al., 2011*). Likewise, the Pcdh19 EC1-4 structure was determined through molecular replacement using Pcdh19 EC3-4 as the initial search model in PHASER. Data collection and refinement statistics are provided in *Table 1*. The final model for Pcdh19 EC3-4 is missing residues 243–246 in chain A, and residues 244–248 in chain B (chains C and D are complete). The Pcdh19 EC1-4 model is missing residues 32–36 in chain A, residue V1 in chain B, and side chains for residues K17, K75, K419 in chain A and for residues K5, R71, and E95 in chain B. All molecular images were generated with VMD (*Humphrey et al., 1996*).

## Differential scanning fluorimetry

The wild-type (WT) and mutant Pcdh19 EC3-4 fragments were purified as described above and used for differential scanning fluorimetry (DSF) (*Niesen et al., 2007*; *Lavinder et al., 2009*). The experiments were repeated three to nine times using protein at 0.3 mg/ml for WT (n = 9), N317S (n = 9), and E290K (n = 3) in buffer (20 mM TrisHCl [pH 8.0], 150 mM NaCl, 2 mM CaCl$_2$ and 1 mM DTT) mixed with SYPRO Orange dye (final concentration 5x; Invitrogen). Fluorescent measurements were performed in a BioRad CFX96 RT-PCR instrument while samples were heated from 10°C to 95°C in 0.2°C steps. Melting temperatures were estimated when the normalized fluorescence reached 0.5.

## Analytical size exclusion chromatography

Refolded proteins (Pcdh19 EC1-4 WT, E313K, and T146R) were separated from unfolded aggregate protein on a Superdex200 16/60 column (GE Healthcare) with 20 mM TrisHCl [pH 8.0], 150 mM NaCl, 2 mM CaCl$_2$ and 1 mM DTT at 4°C. The fraction corresponding to greatest absorbance was run subsequently on a Superdex200 PC3.2/3.0 column with the same buffer at 4°C. An AKTAmicro system provided a controlled flow rate of 50 µl/min with the sample being injected from a 100 µl loop.

## Bead aggregation assays

Bead aggregation assays were modified from those described previously (*Emond and Jontes, 2014*; *Emond et al., 2011*; *Sivasankar et al., 2009*) to detect the weak homophilic adhesion of Pcdh19EC. The Pcdh19ECFc fusion constructs were transfected alone or with NcadEC W2A/R14E His into HEK293 cells using calcium-phosphate transfection (*Kwon and Firestein, 2013*; *Barry et al., 2014*; *Jiang and Chen, 2006*). Briefly, solution A (10 µg of plasmid DNA and 250 mM CaCl$_2$) was added drop-wise to solution B (2x HBS) while mildly vortexing, and the final transfection solution was added drop-wise to two 100 mm dishes of cultured HEK293 cells. The next day, cells were rinsed twice with 1xPBS and serum-free media. Cells were allowed to grow in the serum-free media for 2–3 days before collecting the media containing the secreted Fc fusions. The media was concentrated using ultracel (Millipore) and incubated with 1.5 µl of protein G Dynabeads (Invitrogen) while rotating at 4°C for 1–3 hr. The beads were washed in binding buffer (50 mM TrisHCl [pH 7.5], 100 mM NaCl, 10 mM KCl, and 0.2% BSA) and split into two tubes with either 2 mM EDTA or 2 mM CaCl$_2$. Beads were allowed to aggregate in a glass depression slide in a humidified chamber for 60 min without motion, followed by two 1 min intervals of rocking (five oscillations/min, ±7° from horizontal). Images were collected upon adding EDTA or CaCl$_2$, after 60 min incubation, and after each rocking interval using a microscope (AxioStar; Carl Zeiss) with a 10x objective. Bead aggregates were quantified using ImageJ software as described previously (*Emond et al., 2011*; *Emond and Jontes, 2014*). Briefly, the images were thresholded, the area of the detected aggregate particles was measured in units of pixels, and the average size was calculated. Assays were repeated three times from separate protein preps and their mean aggregate size (± SEM) at each time point was plotted. Assays were excluded from analysis only if western blots failed to show protein expression.

Western blots were performed on a portion of media containing the Fc fusion proteins before incubation with the beads to confirm expression and secretion of the protein. The media was mixed with sample loading dye, boiled for 5 min and loaded onto 10% Bis-Tris NuPAGE gels (Invitrogen) for electrophoresis. Proteins were transferred to PVDF membrane (GE healthcare) and blocked with 5% nonfat milk in TBS with 0.1% tween before incubating overnight with anti-human IgG or anti-His (1:200 Jackson ImmunoResearch Laboratories, Inc.; 1:1000 NeuroMab). After several washes, the blot was incubated with anti-goat or anti-mouse HRP-conjugated secondary (1:5000, Santa Cruz Biotechnology; 1:5000 Jackson ImmunoResearch Laboratories Inc.) for chemiluminescent detection with Western Lightning substrate (Perkin Elmer).

## Pull-down assays

HEK293 cells were transfected with Pcdh19ECFc (wild-type or mutant) and NcadEC W2A/R14E His constructs using calcium-phosphate transfection as described above. Briefly, solution A (8 µg of plasmid DNA, and 250 mM CaCl$_2$) was added drop-wise to solution B (2x HBS) while mildly vortexing, and the final transfection solution was added drop-wise to 60 mm dishes of cultured HEK293 cells.

24 hr after transfection, cells were washed twice with 1x PBS and once with serum free media, then cells were allowed to grow in the serum free media for 2–3 days. Media containing the secreted protein was collected and incubated overnight with 10 µl of protein G dynabeads (Invitrogen) at 4°C. Beads were washed once in wash buffer (20 mM TrisHcl [pH7.5], 150 mM NaCl, 0.5% triton X-100), then re-suspended in loading buffer. In addition, loading buffer was added to a small amount of reserved input media for each sample. The samples were loaded onto 10% Bis-Tris NuPAGE gels (Invitrogen) for electrophoreses. Proteins were transferred to PVDF membrane (GE healthcare) and blocked with 5% nonfat milk in TBS with 0.1% tween before incubating overnight with anti-human IgG or anti-his (1:200 Jackson ImmunoResearch Laboratories, Inc.; 1:1000 NeuroMab). After several washes in TBS with 0.1% tween, the blot was incubated with anti-goat or anti-mouse HRP-conjugated secondary (1:5000, Santa Cruz Biotechnology; 1:5000 Jackson ImmunoResearch Laboratories Inc.), washed, and developed with chemiluminescent detection with Western Lightning substrate (Perkin Elmer).

### Sequence analysis and residue numbering

For analysis of Pcdh19 residue conservation across species, 102 sequences were obtained from the NCBI protein database and processed manually to include only the extracellular domain through the end of EC4, using the canonical calcium-binding motifs and SignalP4.1 (*Petersen et al., 2011*) as guides. These Pcdh19 sequences (*Figure 5—source data 1*) were then aligned using Clustal Omega (*Sievers and Higgins, 2014*) and the alignment file was put into ConSurf (*Ashkenazy et al., 2016*) to calculate relative conservation of each residue and categorize the degree of conservation into nine color bins. Similarly, conservation between selected δ and clustered protocadherins was calculated in ConSurf. All human δ-protocadherin sequences and sequences for deposited structures of clustered protocadherins were selected and aligned to the sequences from our structure (5IU9) for input into Consurf (*Figure 5—source data 2*). Residue numbering throughout the text and in the structure corresponds to the processed protein, except when referencing human disease mutations for which the number follows standard numbering for the human protein, including the signal peptide (see also *Figure 1—source data 1*).

### PCDH19-FE mutation list

The PCDH19 Female Epilepsy (PCDH19-FE) disease has been cataloged in the Online Mendelian Inheritance in Man (OMIM 300088) and has previously been referred to by several different names including: Juberg-Hellman syndrome, epilepsy and mental retardation limited to females (EFMR), and Early Infantile Epileptic Encephalopathy-9 (EIEE9). A thorough list of the currently known PCDH19-FE mutations is presented in *Figure 1—source data 1*.

### Prediction of glycosylation and glycation sites

Potential Pcdh19 glycosylation sites were predicted for both the human (NP_001171809.1) and zebrafish (ACQ72596.1) sequences using the following servers: NetNGlyc 1.0 (N-glycosylation, GlcNAc-β-Asn), NetOGlyc 4.0 (O-glycosylation, GalNAC-α-Ser/Thr) (*Hansen et al., 1998*; *Steentoft et al., 2013*), and NetCGylc 1.0 (C-glycosylation, Man-α-Trp) (*Julenius, 2007*). In addition, we mapped conserved O-mannosylation sites found in the related δ-protocadherins (*Vester-Christensen et al., 2013*), and mapped the glycosylation sites found in published clustered protocadherin structures from mammalian cells (*Rubinstein et al., 2015*). Potential Protocadherin-19 glycation sites were predicted using the NetGlycate 1.0 server for both the human and zebrafish sequences (*Johansen et al., 2006*).

### Accession numbers

Coordinates for Pcdh19 EC1-4 and EC3-4 have been deposited in the Protein Data Bank with entry codes 5IU9 and 5CO1, respectively.

## Acknowledgements

We thank M Emond, D Choudhary, and other members of the Sotomayor and Jontes laboratories for assistance and discussions. This work was supported in part by the Ohio State University and by

NIH (R21MH098463 to JDJ). Use of APS NE-CAT beamlines was supported by NIH (P41 GM103403 and S10 RR029205) and the Department of Energy (DE-AC02-06CH11357) through grant GUP 40277. SRC is a Jeffrey J Seilhamer fellow and MS is an Alfred P Sloan fellow (FR-2015–65794).

## Additional information

### Funding

| Funder | Grant reference number | Author |
|---|---|---|
| National Institutes of Health | R21MH09463 | James D Jontes |
| Ohio State University | | Marcos Sotomayor |
| U.S. Department of Energy | GUP 40277 | Marcos Sotomayor |
| Alfred P. Sloan Foundation | FR-2015-65794 | Marcos Sotomayor |

The funders had no role in study design, data collection and interpretation, or the decision to submit the work for publication.

### Author contributions

SRC, JDJ, MS, Conception and design, Acquisition of data, Analysis and interpretation of data, Drafting or revising the article

### Author ORCIDs

James D Jontes, http://orcid.org/0000-0002-8954-6127
Marcos Sotomayor, http://orcid.org/0000-0002-3333-1805

## Additional files

### Major datasets

The following datasets were generated:

| Author(s) | Year | Dataset title | Dataset URL | Database, license, and accessibility information |
|---|---|---|---|---|
| Cooper SR, Jontes JD, Sotomayor M | 2015 | Crystal Structure of Zebrafish Protocadherin-19 EC3-4 | http://www.rcsb.org/pdb/search/structid-Search.do?structureId=5CO1 | Publicly available at the RCSB Protein Data Bank (accession no: 5CO1) |
| Cooper SR, Jontes JD, Sotomayor M | 2016 | Crystal Structure of Zebrafish Protocadherin-19 EC1-4 | http://www.rcsb.org/pdb/search/structid-Search.do?structureId=5IU9 | Publicly available at the RCSB Protein Data Bank (accession no: 5IU9) |

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
