## [Decision Letter]

Thank you for submitting your article "Structural determinants of adhesion by Protocadherin-19 and implications for its role in epilepsy" for consideration by *eLife*. Your article has been favorably evaluated by Richard Aldrich (Senior Editor) and two reviewers, one of whom is a member of our Board of Reviewing Editors. The reviewers have opted to remain anonymous.

The reviewers have discussed the reviews with one another and the Reviewing Editor has drafted this decision to help you prepare a revised submission.

Summary:

This paper presents a structural analysis of protocadherin-19, a member of the non-clustered protocadherin subfamily, and its homophilic adhesive mechanism. Unlike classical cadherins, the interaction involves antiparallel association of the first 4 cadherin domains (ECs). The authors verify the importance of the crystallographically-observed interface with mutations that ablate homodimer formation, although a negative control with the alternative interface is not presented. They can also rationalize the extensive set of natural mutations associated with the female-limited form of infant-onset epilepsy. Sequence analysis and comparison to the known interfaces of α and β protocadherins indicates that the mode homodimerization observed here, which is very different from that of classical cadherins, is likely to be common to all protocadherins.

Essential revisions:

The reviewers agree that the structural analysis indicates that the antiparallel dimeric arrangement is a general feature of protocadherins, and that this represents a significant contribution to the field. However, given the earlier work of Jontes and colleagues on the need for N-cadherin in biologically relevant adhesion, it is not clear whether the homophilic interface described for PDCH19 here has a biological function. The extensive analysis of FE mutations does not provide any insight into this problem, and the effects of many of them on EC folding or Ca^2+^ ligation could have been predicted from sequence analysis. To strengthen the case that the homophilic interface is in fact relevant to biological adhesion, the authors need to assess if the loss of adhesion in the homodimer interface mutants (Figure 4) is maintained in the presence of the N-cadherin extracellular fragment used in the 2011 paper. Although this would not distinguish whether these affect homophilic adhesion or an allosteric effect of N-cadherin, if adhesion is lost in the mutant in the presence of N-cadherin they would have a strong case for biological relevance of the interface. In this regard, they should discuss how N-cadherin might affect the binding interactions either allosterically or directly.

In the analysis of FE mutations, the T146R and E313K mutants shown in Figure 4 are notably shifted in their migration. Can the authors exclude that the failure of these mutants to aggregate is due to being improperly processed rather than an interface disruption?

[Editors' note: further revisions were requested prior to acceptance, as described below.]

Thank you for resubmitting your work entitled "Structural determinants of adhesion by Protocadherin-19 and implications for its role in epilepsy" for further consideration at *eLife*. Your revised article has been favorably evaluated by Richard Aldrich (Senior Editor), and a Reviewing Editor.

The manuscript has been improved but there are some remaining issues that need to be addressed before acceptance, as outlined below:

The authors performed the requested experiment in which the N-cadherin extracellular domain is added to the bead aggregation assay. The results confirm that the presence of N-cadherin enhances affinity in the wild type case, and appear to show that the interface mutants have the expected effect of weakening (T146R) or ablating (E313K) the homophilic interaction. However, the authors verify only that there are constant amounts of the PDCH19 and N-cad extracellular fragments in the media used in these experiments, but do not verify their binding. Why didn't they co-IP from media? Wouldn't this be the more straightforward verification of the interaction than the co-IP experiment shown in Figure 4—figure supplement 2? That experiment looks at full-length, membrane bound PDCH19 and Ncad in transfected cells. It seems that there is considerable unprocessed N-cad, implying that they are not only seeing interaction of cell-surface molecules but also those still in the secretory pathway. More importantly the proportion of N-cad co-IPed correlates with loss of aggregation in the Figure 4—figure supplement 1 experiment. This could suggest that the mutants are somehow disrupting the N-cad interaction as well. Even though they declined to discuss the effect of N-cadherin in their rebuttal, the authors need to discuss what is happening here.

A few other points need clarification:

1) The use of the N-cad W2A/R14E mutant was explained in the earlier JCB paper from Jontes' group, but it would be helpful to explain this here (i.e. its effect on PDCH19 does not depend on its ability to mediate hemophilic adhesion).

2) In the discussion of potential glycation sites, the authors do not make clear whether glycation of PDCH19 has been documented. Please clarify. If not, this should be removed or made clear that it is speculative.

3) In Figure 1—figure supplement 1, please label the domains and a few of the side chains so that readers can understand what they are looking at.

---

## [Author Response]

*[…] Essential revisions:*

*The reviewers agree that the structural analysis indicates that the antiparallel dimeric arrangement is a general feature of protocadherins, and that this represents a significant contribution to the field. However, given the earlier work of Jontes and colleagues on the need for N-cadherin in biologically relevant adhesion, it is not clear whether the homophilic interface described for PDCH19 here has a biological function. The extensive analysis of FE mutations does not provide any insight into this problem, and the effects of many of them on EC folding or Ca^2+^ ligation could have been predicted from sequence analysis. To strengthen the case that the homophilic interface is in fact relevant to biological adhesion, the authors need to assess if the loss of adhesion in the homodimer interface mutants (Figure 4) is maintained in the presence of the N-cadherin extracellular fragment used in the 2011 paper. Although this would not distinguish whether these affect homophilic adhesion or an allosteric effect of N-cadherin, if adhesion is lost in the mutant in the presence of N-cadherin they would have a strong case for biological relevance of the interface. In this regard, they should discuss how N-cadherin might affect the binding interactions either allosterically or directly.*

In two previous papers we showed that Protocadherin-19 can interact with N-cadherin, and that N-cadherin can facilitate or enhance Protocadherin-19 interactions *in vitro*. However, we never claimed (nor would we), that this constituted the only “biological adhesion”, in that this is the only circumstance under which Protocadherin-19 functions *in vivo*. For example, N-cadherin interacts in *cis* with many proteins, including the FGF receptor, Nectin and protocadherins, as well as forming *cis*-dimers and mediating adhesion in the absence of these interactions; it wouldn’t be argued that only one of these is biologically relevant. The same is almost certainly true for Protocadherin-19 and other protocadherins. Nevertheless, we have repeated the bead aggregation assays in the presence of N-cadherin and find that mutations in the adhesive interface Pcdh19-I1 still impair aggregation (subsection “Binding Assays Probing Pcdh19 Interfaces”, third paragraph; Figure 4—figure supplement 1). New co-immunoprecipitation experiments also show that mutant protocadherin-19 proteins with impaired adhesion still interact with N-cadherin (in the aforementioned paragraph; Figure 4—figure supplement 2). In addition, we now include a negative control. In this control a mutation expected to disrupt a salt-bridge in an alternate crystallographically observed interface does not abolish bead aggregation (in the aforementioned paragraph; Figure 2—figure supplement 2, panels B-D). These additional data, along with results from our original manuscript, suggest that the adhesive interface Pcdh19-I1 of Protocadherin-19 observed in our crystal structure is also the interface used in the presence of N-cadherin. We refrained from speculating about how N-cadherin may affect Protocadherin-19 adhesion (allosterically or directly), as more experiments are needed to fully understand their interaction mode.

*In the analysis of FE mutations, the T146R and E313K mutants shown in Figure 4 are notably shifted in their migration. Can the authors exclude that the failure of these mutants to aggregate is due to being improperly processed rather than an interface disruption?*

We include Figure 6 as an image of the original full SDS-page gel and western blot that shows all mutants running as the wild-type protein. The apparent shift in migration is due to “smiling of the gel,” and placement of lanes 2 and 3 (T146R and E313K mutants) next to lane 6 (wild type). We have changed the order of lanes in the revised manuscript figure to avoid confusion (Figure 4). In addition, we have verified that the mutation E313K does not dramatically affect the stability of protocadherin-19 (subsection “PCDH19-FE Mutations Analyzed in the Context of the Pcdh19 EC1-4 Structure”, last paragraph; Figure 1).

Author response image 1.**DOI:**
http://dx.doi.org/10.7554/eLife.18529.027

[Editors' note: further revisions were requested prior to acceptance, as described below.]

*The manuscript has been improved but there are some remaining issues that need to be addressed before acceptance, as outlined below:*

*The authors performed the requested experiment in which the N-cadherin extracellular domain is added to the bead aggregation assay. The results confirm that the presence of N-cadherin enhances affinity in the wild type case, and appear to show that the interface mutants have the expected effect of weakening (T146R) or ablating (E313K) the homophilic interaction. However, the authors verify only that there are constant amounts of the PDCH19 and N-cad extracellular fragments in the media used in these experiments, but do not verify their binding. Why didn't they co-IP from media? Wouldn't this be the more straightforward verification of the interaction than the co-IP experiment shown in Figure 4—figure supplement 2? That experiment looks at full-length, membrane bound PDCH19 and Ncad in transfected cells. It seems that there is considerable unprocessed N-cad, implying that they are not only seeing interaction of cell-surface molecules but also those still in the secretory pathway. More importantly the proportion of N-cad co-IPed correlates with loss of aggregation in the Figure 4—figure supplement 1 experiment. This could suggest that the mutants are somehow disrupting the N-cad interaction as well. Even though they declined to discuss the effect of N-cadherin in their rebuttal, the authors need to discuss what is happening here.*

We have carried out new pull-down experiments that directly test the interaction between the extracellular domains of Pcdh19 and N-cadherin (Pcdh19ECFc and NcadECHis). These results show that the T146R and E313K mutations in Pcdh19 do not abolish the interaction with N-cadherin, as shown in the new figure Figure 4—figure supplement 2. We note that “unprocessed” N-cadherin (seen in these experiments) is known to reach the cell surface, and this likely happens *in vivo* as well (Latefi et al., Dev. Neurobiol. 2009). As for the proportion of N- cadherin pulled down, it would be inappropriate to use these assays to get a quantitative evaluation of the Pcdh19 interaction with N-cadherin. It is possible that these mutations alter the Pcdh19 interaction with N-cadherin in subtle ways, and we have changed the text to acknowledge this possibility (subsection “Binding Assays Probing Pcdh19 Interfaces”, third paragraph; Discussion and Conclusions section, second paragraph; legend of figure Figure 4—figure supplement 2).

*A few other points need clarification: 1) The use of the N-cad W2A/R14E mutant was explained in the earlier JCB paper from Jontes' group, but it would be helpful to explain this here (i.e. its effect on PDCH19 does not depend on its ability to mediate hemophilic adhesion).*

We have incorporated explanations and references in the main text and figure legends (subsection “Binding Assays Probing Pcdh19 Interfaces”, third paragraph; legend of Figure 4—figure supplement 1).

*2) In the discussion of potential glycation sites, the authors do not make clear whether glycation of PDCH19 has been documented. Please clarify. If not, this should be removed or made clear that it is speculative.*

A note has been added indicating that glycation has never been reported for cadherins (subsection “Antiparallel Interfaces in Crystal Contacts of the Pcdh19 EC1-4 Structure”, fourth paragraph; legend of Figure 2—figure supplement 3).

*3) In Figure 1—figure supplement 1, please label the domains and a few of the side chains so that readers can understand what they are looking at.*

Labels have been added to a revised Figure 1—figure supplement 1.